# Male DAT Val559 Mice Exhibit Compulsive Behavior under Devalued Reward Conditions Accompanied by Cellular and Pharmacological Changes

**DOI:** 10.3390/cells11244059

**Published:** 2022-12-15

**Authors:** Adele Stewart, Gwynne L. Davis, Lorena B. Areal, Maximilian J. Rabil, Vuong Tran, Felix P. Mayer, Randy D. Blakely

**Affiliations:** 1Department of Biomedical Science, Florida Atlantic University, Jupiter, FL 33458, USA; 2Stiles-Nicholson Brain Institute, Florida Atlantic University, Jupiter, FL 33458, USA

**Keywords:** psychiatric disorder, compulsivity, impulsivity, dopamine transporter, animal model, instrumental learning, habit, goal

## Abstract

Identified across multiple psychiatric disorders, the dopamine (DA) transporter (DAT) Ala559Val substitution triggers non-vesicular, anomalous DA efflux (ADE), perturbing DA neurotransmission and behavior. We have shown that DAT Val559 mice display a waiting impulsivity and changes in cognitive performance associated with enhanced reward motivation. Here, utilizing a within-subject, lever-pressing paradigm designed to bias the formation of goal-directed or habitual behavior, we demonstrate that DAT Val559 mice modulate their nose poke behavior appropriately to match context, but demonstrate a perseverative checking behavior. Although DAT Val559 mice display no issues with the cognitive flexibility required to acquire and re-learn a visual pairwise discrimination task, devaluation of reward evoked habitual reward seeking in DAT Val559 mutants in operant tasks regardless of reinforcement schedule. The direct DA agonist apomorphine also elicits locomotor stereotypies in DAT Val559, but not WT mice. Our observation that dendritic spine density is increased in the dorsal medial striatum (DMS) of DAT Val559 mice speaks to an imbalance in striatal circuitry that might underlie the propensity of DAT Val559 mutants to exhibit compulsive behaviors when reward is devalued. Thus, DAT Val559 mice represent a model for dissection of how altered DA signaling perturbs circuits that normally balance habitual and goal-directed behaviors.

## 1. Introduction

Behavioral automaticity, arising from the formation and execution of adaptive habits, ensures efficient and productive use of cognitive processes by ensuring that focus and attention are directed toward salient stimuli. However, across several psychiatric disorders, maladaptive behaviors are or may become habitual and, in extreme cases, lead to the development of compulsions. As such, subjects with attention deficit hyperactivity disorder (ADHD), autism spectrum disorder (ASD), Tourette Syndrome, obsessive compulsive disorder (OCD), substance use disorders (SUDs) display functional neurological deficits that may bias them to rely excessively on striatal-dependent habit memory/learning [1]. The transition between goal-directed and habitual behavior is controlled by midbrain dopaminergic nuclei, the striatum, and cortical regions such as the infralimbic and medial prefrontal cortex, which project to the striatum [2]. Several lines of evidence have pointed to a shift in dominant striatal activity from ventral to dorsal with the progression of habit learning [3] and ultimately from the dorsomedial (DMS) to dorsolateral (DLS) striatum for entrenched habits and compulsions [4,5]. These two striatal subregions are differentially recruited across the acquisition of instrumental learning tasks [5] and may either compete or cooperate to direct action selection as training progresses [6,7]. Lesions to the DLS allow for maintenance of goal-directed behavior but impair habit formation [8] while DMS lesions accelerate the emergence habitual behaviors [9].

The dorsal striatum is densely innervated by dopaminergic projections from the substantia nigra pars compacta (SNc), a neuron population implicated in learning and the retention of habits [10,11]. Habit formation can also be accelerated by exposing animals to drugs that increase dopaminergic tone [12,13]. Here again, however, the exact role of DA in instrumental learning may differ by circuit, with DLS DA linked to cue salience and motivation [14] whereas DA in the DMS facilitates the emergence of compulsive behavior [15,16]. What remains unclear is how psychiatric disease risk factors might influence the dynamic interplay between striatal subregions necessary to achieve appropriate cue-dependent action selection. 

The presynaptic DA transporter (DAT, *SLC6A3*) is responsible for rapid DA clearance in the dorsal striatum [17,18,19,20] and several coding variants in the *SLC6A3* gene have been discovered in individuals with psychiatric diagnoses [21,22,23,24,25,26]. Among these is the Ala559Val substitution identified in two brothers with ADHD [23], two unrelated boys with ASD [27], and a girl with bipolar disorder (BPD) [28]. In vitro, the DAT Val559 mutation supports conspicuous nonvesicular DA release we have termed anomalous DA efflux (ADE) [29] and that, in vivo, leads to constitutive activation of D2-type autoreceptors (D2ARs) that drives inhibition of DA synthesis and release as well as enhanced DAT surface localization [30]. Consequently, DAT Val559 mice exhibit perturbations of multiple spontaneous and drug-stimulated behaviors [31,32]. Notably, DAT Val559 mice display a waiting impulsivity, an inhibitory deficit that may be driven by an enhanced motivational state [33]. In this regard, the ability of DAT Val559 to bias DA release from vesicular to transporter mediated release [30,31] predicts that what appears as enhanced motivation may in actuality be a shift in the balance between goal and habit-directed behavior that can result in behavioral inflexibility, a condition seen in several neuropsychiatric disorders and thought to have a significant dopaminergic component [34,35,36,37,38,39,40]. 

Goal directed and habitual behaviors can be distinguished in instrumental learning tasks in rodents by their differential sensitivity to devaluation [5]. Here, we utilize a combination of operant tasks to assess the balance between goal-directed and habitual behaviors and behavioral flexibility of DAT Val559 mutants. We identify context-dependent compulsive and repetitive behaviors that only emerge in DAT Val559 mice when previously rewarded actions are without consequence. 

## 2. Materials and Methods

### 2.1. Animals

All experiments were performed under a protocol approved by the Institutional Animal Care and Use Committees at Florida Atlantic University. Homozygous DAT Val559 and WT littermate mice used in the study were bred from heterozygous breeders on the hybrid background used in our prior studies [31] (75% 129S6 and 25% C57BL/6J). Unless otherwise noted, males were evaluated in the present study owing to the bias toward male subjects for ADHD diagnoses [41]. Animals were housed on a 12:12 (L:D) cycle. Mice were tested during their active cycle, achieved by raising animals on a reverse light cycle with lights on and off at 3 p.m. and 3 a.m., respectively. Unless otherwise noted, mice were approximately 6–7 weeks old when training for different behavioral assays commenced. For all operant conditioning tasks mice were placed on food restriction one week prior to the start of training. Mice were brought to approximately 85–90% of their baseline weight (weighed every other day). On the fourth and fifth days (still under food restriction), mice were exposed to 33% Vanilla Ensure^®^ Original (Ensure^®^, Abbot Laboratories, Chicago, IL, USA), the reward used for all the operant tasks, for one hour in the home cage. Animals were run under red light, with house lights off in the operant chambers. 

### 2.2. Goal vs. Habit Nose Poke Paradigm

Modified from the methods in Gremel et al. 2013, this assay was utilized to examine goal-directed and habit behavior [5]. This assay relies on the ability of different nose poke schedules to predispose an animal to utilize a goal-directed behavioral strategy (random ratio reinforcement schedule) or support habit formation (random interval reinforcement schedule). Male WT (*n* = 12) and DAT Val559 mice (*n* = 10) underwent two training sessions each day in two distinct contexts. Animals were tested in Med-Associates (St. Albans, VT, USA) operant chambers housed in sound attenuation boxes. The chamber had a house light and nose poke hole on one wall and the opposite wall contained 2 nose poke holes, one on either side of the reward delivery zone. To allow for differentiation of training context, the walls perpendicular to the nose poke hole walls consisted of either clear plastic or black and white vertical striped walls. The paradigm included the following stages:The training schedule consisted of two days of FR1-FT10 (fixed ratio 1-fixed time 10) where the trial started with the delivery of a free reward (Ensure^®^) that the mouse had 5 s to consume upon head entry into the reward delivery zone. The mouse could then earn a reward by making a nose poke into a nose poke hole backlit by LEDs, otherwise a reward was freely delivered every 10 s (FR1-FT10). Mice had to nose poke into an active hole (indicated by a lit LED in the back of the hole) to elicit delivery of a reward. Mice were tested under the FR1-FT10 schedule for 15 min sessions and 15 rewards/session for each training context. Only the holes to the left and right of the reward delivery zone could be active holes. This FR1-FT10 schedule was followed by 2 days of FR1-FT30 (15 min sessions, 15 rewards/session). Mice then underwent CRF training (60 min sessions, 15 rewards/context). Under this protocol mice received a free reward at the beginning of the session, but then had to complete a nose poke on the active hole (indicated by LED) to earn additional rewards. All mice had to reach the criterion of earning 15 rewards within 60 min in both training contexts before moving to the next training stage as a group. To prevent overtraining, mice that obtained this criterion sooner were rested, and total training sessions at CRF 15 were kept equivalent across mice +/− 1 training session.Mice were then run three days on CRF with 30 rewards possible. Mice were then trained on random ratio 10 (RR10; reward was delivered on average every 10 nose pokes with a 0.1 probability that a given nose poke would produce a reward) and random interval 30 s (RI30; reward was delivered upon nose poke on average every 30 s regardless of nose poke vigor) for 3 days. Mice were balanced across training context for which hole was the active hole (i.e., for mouse 1 the active nose poke hole was the left hole in the clear plastic environmental context while the right hole was the active nose poke hole in the striped environmental context, but this condition was reversed for mouse 2). After the RR10 and RI30 training the intensity of the schedules were increased to random ratio 20 (RR20) and random interval 60 (RI60) where animals were given 60 min in each context to earn 30 rewards.Upon completion of RR20/RI60 training mice underwent two days of five-minute non-reinforced probe tests done under devalued (Ensure^®^ given in home cage for 1 h) and valued states (mouse chow given in home cage for 1 h), counterbalanced across days, context, and genotype. Prior to the start of the non-reinforced probe tests, mice were separated into clean cages and allowed access to either mouse chow (valued probe test days) or Ensure^®^ (devalued probe test days) for an hour. The amount of chow and Ensure^®^ consumed in home cage was measured to confirm that there were no consumption differences across genotypes. The full training schedule is shown in Figure 1A.

### 2.3. Devaluation of Progressive Ratio

The protocol and baseline data for this assay can be found in Davis et al. [33]. In this experiment, male WT (*n* = 22) and DAT Val559 (*n* = 21) mice were tested on progressive ratio in a devalued state the day after they were tested on progressive ratio under normal conditions. For devaluation mice were given access to Ensure^®^ in their home cage for one hour prior to testing. This deviates from the devaluation above as mice were not separated into single housing, and trials were reinforced, the mice could still earn Ensure^®^ by completing the nose poke requirements as determined by the progressive ratio program. 

### 2.4. Pairwise Discrimination and Reversal Learning

Male WT (*n* = 14) and DAT Val559 mice (*n* = 15) underwent pairwise discrimination training in operant conditioning chambers (Campden Instruments Model 80614, London, UK) beginning at 5–7 weeks of age. Each chamber contained a reward dispenser and receptacle with sensors and light, a house light, and a touch screen covered by a plastic divider creating two distinct screens for stimulus presentation all enclosed within a sound attenuating chamber. During the habituation phase, mice were placed in the chamber with free access to the 33% vanilla Ensure^®^ reward for a maximum of 30 min or 30 retrievals, whichever came first. Next, mice moved on to the initial touch stage where a visual stimulus was presented on one side of the screen pseudo-randomly for 30 s. Mice receive 3× reward if they touch the displayed stimulus, or, if the 30 s expire without a touch to the stimulus, then a normal reward volume is released. Mice must complete 30 trials in 30 min to meet criteria for advancement. During the “must touch” stage, reward is only released after the mouse touches the illuminated side of the screen with animals progressing after completing 30 trials in 60 min. The “must initiate” stage is identical to the “must touch” stage with the exception that the session begins with a free reward delivery, which the mouse must retrieve to initiate the trial and trigger the stimulus display. The protocol for the “punish incorrect” adds an additional layer of stringency to the “must initiate” stage with touches to the blank screen punished with a time-out period of 5 s with the house lights turned on. A mouse must reach 80% accuracy from 30 trials completed in 60 min for two days in a row to advance. During pairwise discrimination, 2 visual stimuli of equal salience are presented to the mouse. One stimulus is programmed as correct and the other incorrect with correct/incorrect stimuli randomly assigned to each mouse. Correct touches trigger reward delivery, but an incorrect touch triggers a time out period (10 s) and the beginning of a correction trial where the stimuli are presented in the same L-R configuration until the mouse chooses the correct stimulus. Mice must complete 30 trials with 80% accuracy and 6 or less correction trials for advancement. Finally, to assess reversal learning, the pairwise discrimination paradigm is repeated but the rewarded and punished stimuli are reversed. 

### 2.5. Apomorphine-Induced Locomotion

Drug-induced locomotion were measured using Med Associates activity chambers as previously described [31]. The open field chamber (27 × 27 × 20.5) are placed within light- and air-controlled sound-attenuating boxes (64 cm × 45 cm × 42 cm). Locomotion was detected by interruption of infrared beams by the body of the mouse (16 photocells in each horizontal axis located 1 cm above the activity chamber floor, as well as 16 photocells elevated 4 cm above the chamber floor to detect rearing behaviors). Data were collected and analyzed by Med Associates Activity Monitor 7 software. Activity testing was performed with prior habituation wherein, on day 1, animals were placed in the chamber for 30 min to acclimate to the testing environment. 2 days later, animals were again habituated for 30 min, injected with sterile saline (0.9% NaCl) and activity was recorded for 60 min post-injection. On the final day of testing, mice received drug injections following an initial 30 min habitation and activity was again monitored for 60 min post-injection. Apomorphine HCl hemihydrate (Sigma, 5 mg/kg, St. Louis, MO, USA) was dissolved in sterile saline and administered subcutaneously (s.c.). Quinpirole hydrochloride (Sigma, 1 mg/kg) was dissolved in sterile saline and administered via intraperitoneal (i.p.) injection. 

### 2.6. Golgi Staining and Dendritic Morphology

Following phosphate-buffered saline (PBS) perfusion, animals were sacrificed by rapid decapitation, and brains from WT (*n* = 4) and DAT Val559 (*n* = 4) male mice were harvested and stained using the FD Rapid GolgiStain kit (FD Neurotechnologies, cat # PK401, Columbia, MD, USA) per the manufacturer’s instructions [42,43]. 100 μm brain sections were imaged on a BZX-800 Keyence (Osaka, Japan) microscope using a 60× objective on brightfield mode. 5 z-stack images (step size of 0.5 µm, 75 images) were taken for each region per mouse. Neuronal tracing was performed using Neurolucida 360 (MBF Bioscience, Williston, VT, USA) for spine density quantification. Dendritic spines were further classified into four types, namely stubby, mushroom, thin, and filopodial extensions, based on their length and head diameter as described in previous studies [44,45].

### 2.7. Statistical Analyses

Statistical analyses was performed using the GraphPad Prism 9 software package (San Diego, CA, USA). Statistical significance was set at *p* ˂ 0.05 for all experimental results. Datasets comparing two independent groups were analyzed by a two-tailed Student’s *t*-test. Datasets with multiple independent variables (e.g., genotype and drug) were analyzed by two-way ANOVA with Bonferroni post hoc tests for individual comparisons. Time course datasets were analyzed by two-way repeated measures (RM) ANOVA. Data are presented as mean ± S.E.M. Complete details of the statistical analysis including effect size for all datasets is included in Appendix A.

## 3. Results

### 3.1. DAT VAL559 Males Display Compulsive Reward Seeking in Devalued States for Both Goal-Directed and Habit Driven Behavioral Contexts

To evaluate goal-directed and habit behavior in DAT Val559 mice, we employed a within subject lever pressing paradigm that utilizes contextual cues coupled to random ratio (RR) or random interval (RI) reinforcement schedules to bias animals toward goal-directed or habitual actions, respectively (Figure 1B) [5]. Following completion of the training paradigm (Figure 1A), both WT and DAT Val559 mice reached stable response rates in both the RR (Appendix A) and RI (Appendix A) contexts. However, unlike results previously shown in C57Bl/J6 mice [5], our WT animals (75% 129S6/SvEvTac, 25% C57Bl/J6) were sensitive to devaluation in both the goal-directed (RR) and habit (RI) context, though the sensitivity of the goal-directed (RR) reinforcement schedule was greater (Figure 1C). DAT Val559 mice also displayed sensitivity to devaluation in the goal-directed context but were more resistant than WT animals to devaluation in the habit context (Figure 1C). Intriguingly, regardless of context, DAT Val559 mice invariably displayed increased head entries into the reward dispenser after devaluation as compared to WT mice (Figure 1D). That is, while they stopped nose poking to obtain reward, DAT Val559 mutants still checked for reward delivery. Importantly, no consumption differences in home cage were seen for either chow or Ensure^®^ between genotypes (Figure 1E). 

### 3.2. DAT Val559 Dysregulated Reward Seeking Is also Present with Progressive Ratio Testing under Devalued Conditions

Next, we were interested in establishing if this compulsive head entry phenotype was unique to the reinforcement schedules (RR, RI) utilized to drive goal-directed or habitual responding. We had previously demonstrated that, under valued conditions, DAT Val559 males display a higher breakpoint in a progressive ratio (PR) task [33]. Similar to results obtained in the goal-directed and habit contexts, while devaluation normalized PR breakpoint to WT levels in DAT Val559 mice (Figure 2A), the mutants still displayed increased reward seeking behaviors as reflected by an increase in head entries to the reward delivery magazine (Figure 2B). Similarly, whereas the rate of correct nose pokes did not differ between genotypes (Figure 2D), DAT Val559 displayed an overall longer session length as compared to WT mice (Figure 2C). 

### 3.3. DAT Val559 Males Display Enhanced Learning in Pairwise Discrimination Task but No Deficits in Cognitive Flexibility

Next, we sought to establish whether the compulsive reward seeking behavior of DAT Val559 mice could be linked to changes in cognitive flexibility or the ability to “unlearn” cue-response associations. To test this hypothesis, we employed a pairwise discrimination paradigm in which animals were trained to choose between two images (Figure 3A), one associated with reward delivery. Though stage progression did not differ significantly between genotypes (Figure 3B), DAT Val559 mice displayed increased accuracy during acquisition of pairwise discrimination (Figure 3C; F(1, 29) = 8.089, *p* = 0.0081) and an overall decrease in the number of correction trials required to progress through the learning protocol (Figure 3D) as compared to their WT littermates. However, the performance of DAT Val559 mice during reversal learning was indistinguishable from that of WT mice (Figure 3E,F and Appendix A). These data indicate that DAT Val559 mice do not display cognitive deficits for this rewarded task.

### 3.4. DAT Val559 Males Exhibit Increased Sensitivity to DA Agonist-Induced Repetitive Motor Behaviors

Our prior analysis of DAT Val559 mice indicated that, in the absence of reward, DAT Val559 mice display perseverative behaviors. For example, when the Y maze spontaneous alternation task was performed in an un-baited manner, DAT Val559 males display decreased alternation and increased direct revisits into previously explored arms of the maze, a phenotype reversible via D2-type receptor (D2R) blockade [46]. Thus, we hypothesized that DAT Val559 mice would display repetitive phenotypes after administration of direct DA agonists. The dopamine D2R/D3R agonist quinpirole stimulated rapid and robust locomotor suppression in mice regardless of genotype (Appendix A), an effect we suspect can be attributed to targeting of pre-synaptic autoreceptors. However, DAT Val559 males displayed a significant increase compared to WT mice in locomotor stereotypies in response to the mixed DA agonist apomorphine (Figure 4C), without an impact on horizontal (Figure 4A) or vertical locomotor activity at the 5 mg/kg dose employed (Figure 4B). DAT Val559 mice of both sexes increase center occupancy in response to apomorphine to a degree comparable to their WT littermates (Appendix A). However, WT females, but not males, increase their vertical exploration following apomorphine treatment, a phenotype lost in DAT Val559 females (Appendix A). Similarly, while there is a trend to increased apomorphine-induced stereotypic behavior in DAT Val559 females compared to WT controls (Appendix A), this difference failed to reach statistical significance (*p* = 0.068).

### 3.5. Dendritic Spine Density Is Increased in the DMS of Male DAT Val559 Mice

Given evidence of baseline and drug-stimulated compulsive behaviors in DAT Val559 mice and the ability of DA to modulate spine density of striatal medium spiny neurons (MSNs) [47], we hypothesized that DAT Val559-driven ADE might drive neuroadaptations in key brain regions linked to compulsions. Our prior studies demonstrated that perturbations in DAT trafficking and function were confined in DAT Val559 males to the dorsal striatum [30]. Given that DA elevations following administration of the psychostimulant cocaine are associated with elevated MSN dendritic spine density [44,48,49,50], we hypothesized that DAT Val559-driven ADE might alter synaptic strength in the striatum. Here, we further parse striatal subregions looking for alterations in spine densities in the DMS and DLS of WT and DAT Val559 mice. Intriguingly, we detected an increase in spine density in the DMS (Figure 5A), but not DLS (Figure 5B) of DAT Val559 males as compared to WT mice. This increase could not be attributed to a specific dendritic morphology (e.g., thin, stubby, or mushroom shaped spines). Further analysis by dendritic branch order, a measure of degree of branching beginning at the cell soma, showed no significant effect of branch ordering in the DMS (F(4, 20) = 0.8154, *p* = 0.5303), though a significant genotype effect was confirmed (F(1, 20) = 7.600, *p* = 0.0122) (Appendix A). 

## 4. Discussion

The balance between goal-directed, purposeful behaviors, and the formation of automatic habits is governed by a dynamic interplay between striatal subregions known to be innervated by dopaminergic projections from midbrain nuclei [51]. Cue-directed behaviors rely on phasic DA release [34,35]. However, in addition to the fast pulse responses observed in reward control regions such as the ventral striatum, DA also shapes behavioral output by encoding uncertainty, punishment, motivation, and movement through engagement of distinct striatal and cortical circuits. Increasing dopaminergic tone, for example, in patients with Parkinson’s disease undergoing therapy with the DA precursor L-DOPA, tips the balance toward habit formation and the development of compulsions [52]. As DAT Val559 mice display tonic ADE, we hypothesized that altered extracellular DA could alter the signal to noise ratio (phasic DA: tonic DA) necessary to establish cue salience required for goal-directed behavior but that also might predispose DAT Val559 mice to shift toward habitual behaviors more readily. Indeed, we observed that, while WT mice displayed a significant drop in nose poke behavior in the RI context following devaluation, DAT Val559 mice retained their nose poke behavior. Further, even when the effort required to work for reward was suppressed, DAT Val559 mice displayed a perseverative reward seeking behavior returning to the reward delivery magazine in the absence of a proper stimulus response. 

Our prior work with the DAT Val559 model demonstrated that these mice exist in a state of enhanced motivation that drives accelerated learning and increased performance under uncertain conditions [33]. In pairwise discrimination and reversal learning, DAT Val559 mice displayed accelerated learning and no issues with the cognitive flexibility necessary to inhibit a prior reinforced behavior and acquire a new stimulus-response association. Thus, we suspect that, for DAT Val559 mice, cognitive deficits may be masked by an increased motivation for reward acquisition. Indeed, in the 5-choice serial reaction time task (5-CSRTT), it was only when DAT Val559 mice were expected to wait for stimulus presentation that they began to exhibit an increase in premature responding [33]. Similarly, in the devalued state, irrespective of reinforcement schedule, DAT Val559 mice display no difference from WT animals in effort expended (e.g., active nose pokes), but rather displayed a compulsive reward seeking behavior, returning to the reward delivery magazine even when they have not worked to obtain reward. This unique phenotype is likely not due to a difference in satiation levels as no difference in home cage consumption of chow or Ensure^®^ was observed between genotypes. It is unclear whether the elevated head entries in the DAT Val559 mouse represent a compulsion to check for reward. Regardless, perseveration and entrenched stimulus-response associations in the absence of reward are characteristic of male DAT Val559 mice. For example, in the unrewarded Y Maze, DAT Val559 males display a decrease in spontaneous alteration and an increase in direct revisits to the same arm of the maze. Similarly, DAT Val559 males display delayed extinction (elimination of the drug-context association) in a cocaine-induced conditioned place preference paradigm [32]. These observations are consistent with maladaptive habit memory formation that emerges when the presumed motivational drive elicited by a salient reward stimulus is removed.

Prior work in the DAT Val559 model identified a region-specific, ADE-driven feedback onto presynaptic D2ARs that drives aberrant phosphorylation and trafficking of mutant DATs, with selectively for the dorsal striatum of male mice [30]. There is a substantial body of literature indicating that chronic administration of the D2R/D3R agonist quinpirole can induce compulsive checking behaviors in rodents [53]. Notably, in an operant task, quinpirole triggered perseverative pressing of a lever, that, when pushed, provided no reward but signaled which lever would be rewarded in a subsequent trial. This phenotype could be mitigated by administration of the D2R antagonist sulpiride [54]. Intriguingly, compulsive checking behaviors in the chronic quinpirole model can also be mitigated by providing rats with an untreated novel social stimulus [55] indicating that activation of brain circuits responsive to social reward can overcome the tendency of these animals toward perseverative behaviors. We noted that although the acute effects of quinpirole were unaltered in DAT Val559 mice, possibly due to the dominance of autoreceptor-dependent hypoactivity at this dose (1 mg/kg, i.p.) and time course (2 h), we did note an increased sensitivity to apomorphine-dependent motor stereotypies in DAT Val559 mutants. Apomorphine is a mixed DA partial agonist with approximately 10-fold selectivity for D2R/D3Rs over D1Rs. As DAT Val559 mice display decreased sensitivity to D1 agonists, these data might speak to altered activity of D2R/D3Rs in DAT Val559 mice though apomorphine also antagonizes several serotonin and norepinephrine receptors [56]. Additional insight into the role of D2Rs in habitual behaviors has been gleaned from the study of D2R knockout mice, which display a deficit in reversal learning [57] and increased premature responding in the 5-CSRTT [58]. Specific deletion of D2Rs from dopaminergic neurons (D2ARs), conversely, failed to significantly impact reversal learning, but did impair the ability of mice to sustain a prolonged nose poke response [59]. Together these data speak to a role for post-synaptic D2Rs in habitual behaviors and pre-synaptic D2ARs in motivation to work for reward. Due to the existence of multiple populations of D2Rs, parsing out the contribution of each receptor population to phenotypes observed in DAT Val559 mutants will require incorporation of genetic tools in addition to pharmacological manipulations. 

The observation that chronic quinpirole is necessary to trigger compulsive checking in rodents suggests that this phenotype requires long-term neural adaptations to repeated D2R/D3R stimulation. Indeed, administration of quinpirole during adolescence decreases spine formation in the hippocampus, while the D2R antagonist eticlopride has the opposite effect [60]. Similarly, the hyperdopaminergia resulting from knockout of DAT also leads to loss of spines in striatal medium spiny neurons (MSNs) [61] and overexpression of the neuronal glutamate transporter EAAT3 in dopamine neurons during development, but not adulthood, triggers changes in dopamine neuron activity and stereotypic behavior [62]. Given the critical role of the dorsal striatum in habitual behavior, we probed for alterations in dendritic spines in striatal subregions, identifying an increase in spine density specific to the DMS. Intriguingly, the directionality of impacts on spine density elicited by the DAT Val559 mutation are opposite to that observed via either chronic quinpirole or DAT knockout [60,61]. This may be due to selective penetrance of the DAT Val559 driven ADE in the dorsal striatum [30] vs. more global changes in extracellular DA homeostasis that would result from a constitutive knockout. Prior work has demonstrated that activity in the DLS and DMS exist in a delicate balance such that perturbations in either region will result in emergent dominance of the other and a corresponding shift in behavioral strategies [5]. Intriguingly, lesions to the DMS have been shown to maintain elevated head entry under devalued conditions, but not independent of effort (e.g., nose pokes) as we observe in DAT Val559 mice. In addition, while DA release in the ventral striatum and DLS remains stable throughout habit development, DA dynamics in the DMS differ significantly in habitual and non-habitual rats across action-sequence initiation and completion [16]. Thus, alterations in synaptic strength between the DMS and DLS might explain the bias of DAT Val559 toward compulsive behaviors. We should note that the analysis included in this report failed to subclassify MSNs into D1- or D2-types, which are differentially sensitive alterations in synaptic strength elicited by cocaine [44]. Consideration of neuronal subtype impact on spine density is particularly called for as an overall density change was detected but no specific class of spines shows a contribution, which could reflect plasticity for both D1 and D2 populations but with different classes altered, thereby washing out subgroup effects. However, we presently speculate that ADE-driven increases in spine density may be confined to D2-type MSNs given the decreased sensitivity of DAT Val559 mice to D1R agonists and our prior observation indicating that the decrease in spontaneous alternations in the Y maze observed in DAT Val559 males can be reversed via D2R blockade [46]. It is also possible, however, that the phenotypes described here are due to impacts of chronic ADE on other circuits or neurotransmitter systems. For example, selective serotonin (5-HT) reuptake inhibitors have long been first line therapy for OCD and 5-HT bioavailability influences neuronal activity in the dorsal striatum [63,64]. Our prior work demonstrated that altered serotonergic signaling underlies loss of cocaine-dependent hyperlocomotion in DAT Val559 mice [32] providing an alternate hypothesis as to the mechanism underlying the predilection of DAT Val559 mice toward compulsive behaviors. 

It is important to note that our data suggest a key role for the reward circuitry as a sort of override to dorsal striatal dysfunction as the tendency of DAT Val559 mice toward perseveration is absent during rewarded tasks and emerges only when reward is devalued (e.g., all operant tasks), withheld (e.g., CPP extinction) [32], or absent (e.g., Y maze) [46]. These data are consistent with recent evidence from non-human primates whereby the ventral striatum was shown to encode reward value during habit learning, with activity in this region remaining high following presentation of previously rewarded objects even when no reward outcome is expected [65]. Thus, we hypothesize that the enhanced motivational state of DAT Val559 mice might reflect increased DA transmission in the ventral striatum that predominates if reward-associated stimuli are present. When motivation for reward is absent, the dorsal striatum emerges as the prevailing circuit with a bias toward habit formation, and with changes in relative synaptic strength arising in the DMS vs. DLS due to the DAT Val559 mutation. Emerging evidence suggests that engagement of striatal circuitry during operant learning differs by sex [66]. Intriguingly, the impacts of ADE in DAT Val559 mice also differ in a sex- and region-specific manner with DAT Val559 females displaying alterations in DA dynamics specific to the ventral striatum [46]. Thus, it remains to be determined if the phenotypes described here in DAT Val559 males would also be observed in females. Indeed, we observed sex-specific impacts of the DAT Val559 mutation following apomorphine exposure with male DAT Val559 mice displaying increased stereotypic behavior while females display a loss of apomorphine-dependent vertical exploration. These data allude to the possibility that the phenotypes observed in DAT Val559 males might be absent or distinct in DAT Val559 females as we recently demonstrated in other behavioral assays [45]. 

Altered reward processing has been detected in individuals with multiple neuropsychiatric disorders [67,68,69]. While these clinical observations are often attributed to altered reward processing in the ventral striatum, our data in the DAT Val559 model speak to the possibility that, under certain circumstances, increased motivation for reward might be exploited to overcome deficits in cognitive flexibility or a tendency toward compulsive behaviors resulting from imbalances in other striatal subregions. Indeed, performance-based rewards have been shown to improve cognition in patients with both ADHD and ASD by aiding in the identification of salient stimuli and increasing error evaluation [70,71]. As one might predict, the nature of the reward (e.g., monetary vs. social) differentially impacted neural responsiveness in individuals with ASD vs. ADHD [72] and the net effect of reward is often highly individualized and most impactful when personalized to patient interests [73]. DAT Val559 mice represent a model to determine how the dynamic interplay between striatal regions controls salience, motivation, and learning/cognition in a construct-valid model of DA dysfunction. Dissection of the molecular and circuit level changes driving these unique behavioral alterations present in the DAT Val559 mice could provide insights into treatment options for multiple psychiatric disorders for which impulsive and compulsive behaviors contribute to disease presentation. 

## Figures and Tables

**Figure 1 cells-11-04059-f001:**
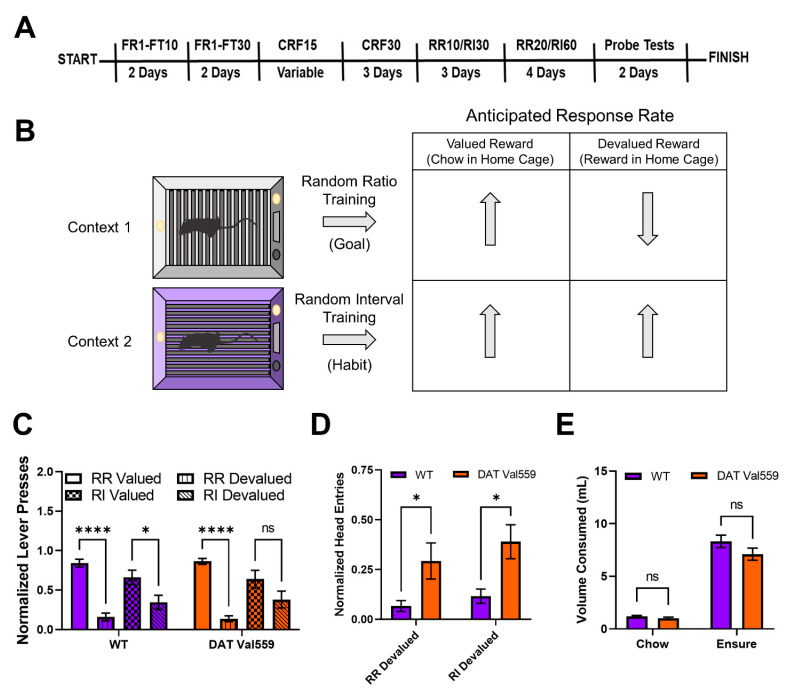
DAT Val559 males display compulsive reward seeking in devalued states for both goal-directed and habit driven behavioral contexts. WT (*n* = 12) and DAT Val559 (*n* = 10) males were subjected to a within subject lever pressing paradigm that utilizes contextual cues coupled to random ratio (RR) or random interval (RI) reinforcement schedules to bias animals toward goal-directed or habitual actions, respectively. (**A**) Overview of the training paradigm. (**B**) Representation of outcome expectations based on training schedule. (**C**) Normalized lever presses in the RR valued, RR devalued, RI valued, and RI devalued contexts. Two-way ANOVA revealed a significant impact of schedule/valuation (F(3, 80) = 32.32, *p* < 0.0001). (**D**) Normalized head entries into the reward delivery magazine. Two-way ANOVA revealed a significant effect of genotype (F(1, 37) = 14.82, *p* = 0.0005). (**E**) Reward (Ensure^®^) or chow consumption in the home cage prior to testing. No significant effects of genotype on reward or chow consumption were detected (F(1, 40) = 2.661, *p* = 0.1107). Data were analyzed by two-way ANOVA with Sidak’s multiple comparison’s test. * *p* < 0.05, **** *p* < 0.0001. ns = not significant. Data are presented as mean ± SEM.

**Figure 2 cells-11-04059-f002:**
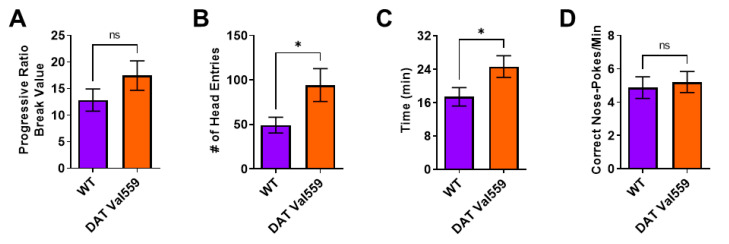
DAT Val559 males show elevated reward seeking phenotypes during devalued progressive ratio. The progressive ratio paradigm was performed in WT (*n* = 22) and DAT Val559 (*n* = 21) males following reward devaluation. (**A**) Progressive ratio break point. (**B**) Head entries into the reward delivery magazine. (**C**) Experimental session length. (**D**) Response rate. Data were analyzed by two-tailed Student’s *t*-test. * *p* < 0.05. ns = not significant. Data are presented as mean ± SEM.

**Figure 3 cells-11-04059-f003:**
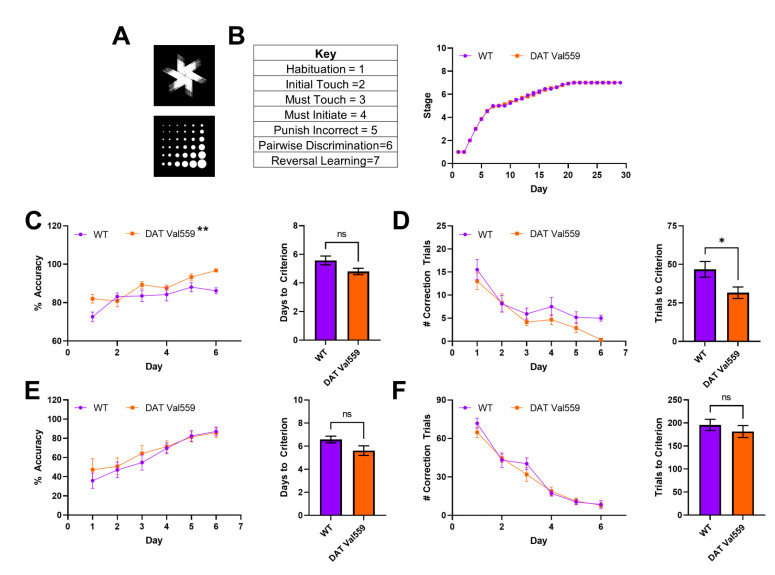
Reversal learning is intact in male DAT Val559 mice. WT (*n* = 14) and DAT Val559 (*n* = 15) males underwent pairwise discrimination training followed by a reversal phase. The two stimulus images are depicted in (**A**). (**B**) Stage progression throughout the duration of the training paradigm. (**C**) % Accuracy during pairwise discrimination (left) and days to reach accuracy criterion (right). Two-way RM ANOVA revealed a significant effect of genotype (F (1, 29) = 8.089, *p* = 0.0081, ** on graph). (**D**) Correction trials by day (left) and total correction trials (right) during pairwise discrimination. (**E**) % Accuracy during reversal learning (left) and days to reach accuracy criterion (right). (**F**) Correction trials by day (left) and total correction trials (right) during pairwise discrimination. Note that, for this experiment, only a few animals (<5/group) took longer than 6 days to progress. As a result, days 7–10 are not included in the graphs due to the potential for the low N to artificially inflate genotype effects. Data were analyzed by two-tailed Student’s *t*-test or two-way RM-ANOVA with Sidak’s multiple comparison’s test. * *p* < 0.05, ** *p* < 0.01. ns = not significant. Data are presented as mean ± SEM.

**Figure 4 cells-11-04059-f004:**
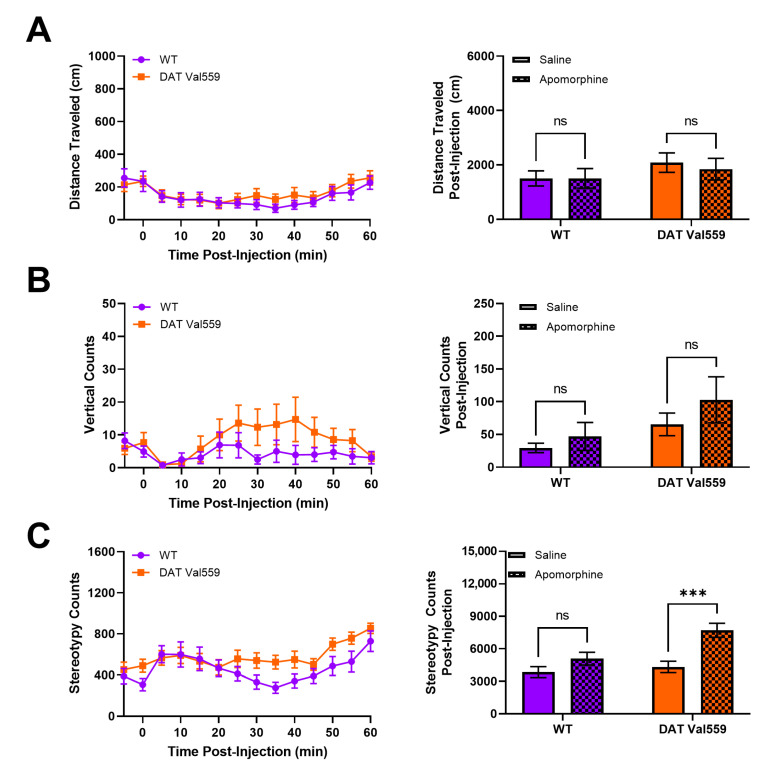
DAT Val559 males exhibit enhanced sensitivity to apomorphine-dependent repetitive motor movements. WT (*n* = 13) and DAT Val559 (*n* = 15) mice were given a single injection of the DA agonist apomorphine (5 mg/kg, s.c.) and locomotor activity recorded for 60 min post-injection. Datasets are presented in 5 min time bins across the recording period and as summary data adding up all activity post-injection. (**A**) Horizontal distance traveled. (**B**) Vertical locomotor activity. (**C**) Stereotypic motor movements. Data were analyzed by two-way ANOVA with Sidak’s multiple comparison’s test. *** *p* < 0.001. ns = not significant. Data are presented as mean ± SEM.

**Figure 5 cells-11-04059-f005:**
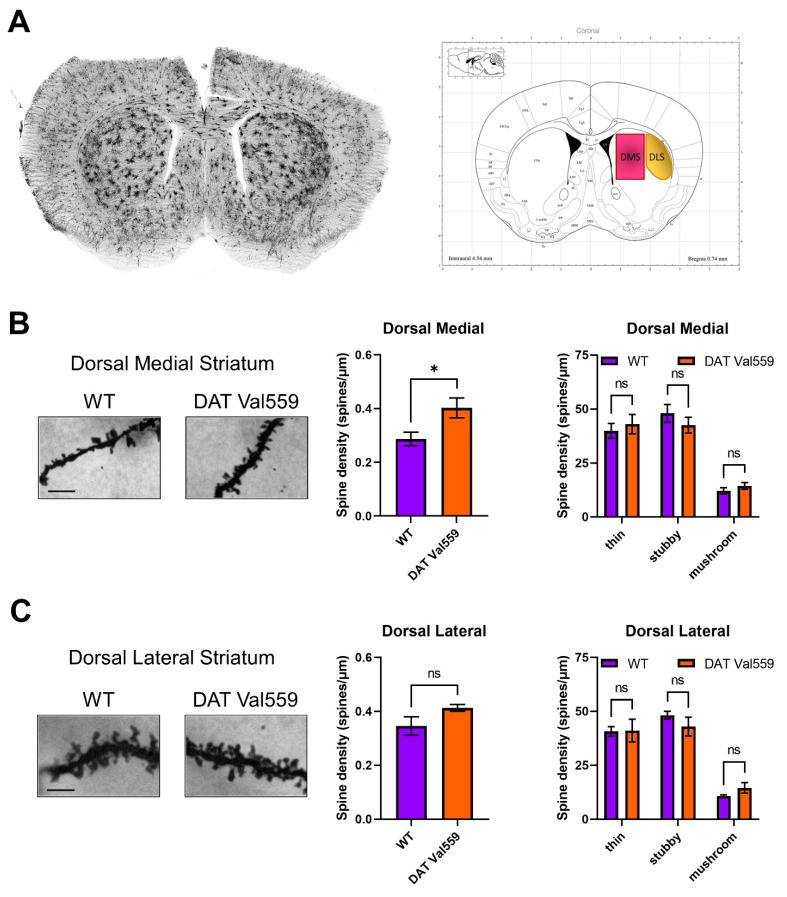
Dendritic spine density is increased in the DMS of male DAT Val559 mice. Spine density in sections from WT (*n* = 4) and DAT Val559 (*n* = 4) mice was assessed utilizing Golgi staining coupled to brightfield microscopy. (**A**) Representative whole coronal slice and notation of regions analyzed. Representative images [scale bar = 100 μm], total spine density, and spine densities for each distinct morphology (thin, stubby, and mushroom) are provided for both the (**B**) DMS and (**C**) DLS. Data were analyzed by two-tailed Student’s *t*-test or two-way ANOVA with Sidak’s multiple comparison’s test. * *p* < 0.05. ns = not significant. Data are presented as mean ± SEM.

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
