# Peer review of "Male DAT Val559 Mice Exhibit Compulsive Behavior under Devalued Reward Conditions Accompanied by Cellular and Pharmacological Changes"

_cells, 2022, doi:10.3390/cells11244059_

Round 1

Reviewer 1 Report

I have reviewed the manuscript by Adele Stewart and coworkers titled “Compulsive Behavior Under Devalued Reward Conditions Ac-companied by Alterations in Cellular and Dopamine Receptor Agonist Sensitivity in DAT Val559 Mice”. The authors reference that the Ala559Val substitution has been found in siblings with ADHD, two boys with ASD and a girl with bipolar disorder. In addition, because the  in vitro data also shows that the DAT VAL559 mutations provide conspicuous DA release the authors investigate the balance between goal-directed and habitual behaviors as well as behavioral flexibility in mutant male mice carrying the Ala559Val substitution.

The article is well written and the results are discussed properly. My main concern is why weren’t the experiments also performed on female mice. Furthermore, the authors in the discussion section highlight (lines 451-455) the importance of determining whether the reported results are also present in females.

 Below I list some minor comments:

   (1)      Please move figure 1 to the materials and methods section so that the protocol carried out by the authors is easier to follow. Also please enlarge in figure 1 C the boxes that indicate the experimental conditions in the graph. They are too small to differ (RR valued, RI valued; RR devalued, RI devalued).

   (2)      Please move figure 3 after section 3.3. and figure 4 after section 3.4. 

    (3)      Section 3.4 line 293. Please start the sentence indicating that the given example was performed by another study. The way the example is given seems that the authors performed the Y-maze test. Another alternative is to move this reference to the discussion section.

    (4)      The description of how the open field task was performed is missing from the methodology section. Please add the corresponding description.

(5) The title of the manuscript must indicate that the results were only performed in male mice:

Compulsive Behavior Under Devalued Reward Conditions Accompanied by Alterations in Cellular and Dopamine Receptor Agonist Sensitivity in DAT Val559 male Mice”

Author Response

Reviewer 1:

We thank the reviewer for their comments and suggestions. In response to this reviewer, we have modified figures (Fig. 1), added additional data (Fig. S5), and edited the manuscript to address this reviewer’s concerns.

Our point-by-point response to the reviewer is summarized below:

Comment #1: My main concern is why weren’t the experiments also performed on female mice. Furthermore, the authors in the discussion section highlight (lines 451-455) the importance of determining whether the reported results are also present in females.

We wholeheartedly agree that all the experiments described in this manuscript should be repeated in female mice and have plans to do this in the future. We originally targeted in this study (and the grant that supported it) studies in male mice due to the male bias of ADHD and ASD. Recently, we published a striking alteration in the penetrance of the DAT Val559 mutant with a shift from dorsal striatal-mediated impact to mesocortical impact, which we can see at biochemical, physiological, and neuroanatomical plasticity levels (Stewart et al, Mol Psychiatry 2022, PMID: 36117213). The latter work reveals a broad trait restructuring, within which an extension of the studies of this submission to females should be considered and accompanied by further studies (in both males and females). We hope that the reviewer understands that our motivation in this direction is sincere and that given the time needed to pursue a more complete assessment, we choose to pursue such efforts in a follow-up study.

However, to provide evidence that such a future effort is justified beyond within the current report, we have added a piece of data to the supplement (Fig. S5) depicting the response of WT and DAT Val559 females to apomorphine. In support of the need to perform experiments in both sexes, some phenotypes differ between DAT Val559 males and females. DAT Val559 mice of both sexes increase center occupancy to a degree comparable to their WT littermates. However, WT females, but not males, increase their vertical exploration following apomorphine treatment, a phenotype lost in DAT Val559 females. Similarly, while there is a trend to increased apomorphine-induced stereotypic behavior in DAT Val559 females, this difference failed to reach statistical significance (P = 0.068). These data are now also noted in the discussion.

Comment #2: Please move figure 1 to the materials and methods section so that the protocol carried out by the authors is easier to follow. Also please enlarge in figure 1 C the boxes that indicate the experimental conditions in the graph. They are too small to differ (RR valued, RI valued; RR devalued, RI devalued).

Done. We have made these changes as requested by the reviewer.

Comment #3: Please move figure 3 after section 3.3. and figure 4 after section 3.4. 

Done. We have made these changes as requested by the reviewer.

Comment #4: Section 3.4 line 293. Please start the sentence by indicating that the given example was performed by another study. The way the example is given seems that the authors performed the Y-maze test. Another alternative is to move this reference to the discussion section.

Done. We have made these changes as requested by the reviewer.

Comment #5: The description of how the open field task was performed is missing from the methodology section. Please add the corresponding description.

Additional information on the drug-induced locomotor analysis has been added to section 2.5 as requested.

Comment #6: The title of the manuscript must indicate that the results were only performed in male mice.

Done. We have altered the title as suggested.

Reviewer 2 Report

The MS concerns interesting topics but needs revisions before publication. Followings are points for revisions.

Method: Method 2.2 is hard to follow. The operant chamber looks like a chamber with a retractable single chamber. These details must be described.

Statistical analysis: Effective size should be given.

Data analysis: In analysis of schedule-controlled behavior, there are two phases of behavior namely that in the transient state and that in the steady state.  Usually analysis of the latter behavior is used for analysis of the schedule-controlled behavior. Three or four sessions is too short to obtain the steady-state. The authors can analyze change during three three/four sessions to argue that they analyzed the steady state.

Results:  In figure 3, the authors wrote “a few animals took longer than 6 days to progress were discarded”. Please give exact number of discarded animals.

Histology: Method of scarifying animals must be given. I recommend providing images of sections used for Golgi analysis.

Author Response

We thank the reviewer for their comments and suggestions. In response to this reviewer, we have modified figures (Fig. 5), added additional information (Table S1, S2), added a figure (Fig. S1) and edited the manuscript to address this reviewer’s concerns.

Comment #1: Method: Method 2.2 is hard to follow. The operant chamber looks like a chamber with a retractable single chamber. These details must be described.

We have provided additional details regarding the operant chambers to section 2.2. In addition, we have split section 2.2 into subsections to better describe each element of the experimental design.

Comment #2: Statistical analysis: Effective size should be given.

Complete details of the statistical analysis including effect size for all datasets is now included in Tables S1 and S2.

Comment #3: Data analysis: In analysis of schedule-controlled behavior, there are two phases of behavior namely that in the transient state and that in the steady state.  Usually, analysis of the latter behavior is used for analysis of the schedule-controlled behavior. Three or four sessions is too short to obtain a steady state. The authors can analyze change during three three/four sessions to argue that they analyzed the steady state.

We understand the reviewer’s comment but wish to note that unlike standard schedule-dependent behavioral paradigms, reaching a “steady state” response rate is not an essential feature of the Goal vs Habit Nose poke Paradigm. In fact, extensive overtraining for some schedules will recruit neurocircuitry involved in habitual learning and preclude the switch between goal-directed and habitual action in a within subject manner that represents a key component of this task1. Further, for this task, we adhered to a previously published protocol designed to probe the relative bias between goal-directed vs habitual responding2. In this prior report, response rate accelerated as animals acquired the task (e.g., moved from CRF to RR10/RI30 and finally RR20/RI60) but stabilized on the final two days of training2. In addition, mice earned similar numbers of rewards, earned rewards at a similar rate and made a similar number of head entries into the food port between RI and RR schedule training2. To accommodate the reviewer’s concern, however, we now provide cumulative nose poke numbers across the final days of training (RR20/RI60) prior to the probe tests. Fig. S1 demonstrates that instrumental responding was stable across this phase of the training paradigm in both genotypes (WT & DAT Val559) and contexts: RR (Fig. S1A) and RI (Fig. S1B).

Comment #4: Results:  In figure 3, the authors wrote “a few animals took longer than 6 days to progress were discarded”. Please give the exact number of discarded animals.

We apologize for the confusion on this point. No animals were discarded from the analysis. We just did not include the data from days 7-10 in the graphs as only a few mice (<30 %) took more than 6 days to reach criterion. We have modified the figure legend to clarify this point.

Comment #5: Histology: Method of scarifying animals must be given. I recommend providing images of sections used for Golgi analysis.

We now provide an example slice and image depicting the regions selected for the Golgi analysis. In addition, the methods section now notes that animals were sacrificed by rapid decapitation following PBS perfusion. 

Round 2

Reviewer 1 Report

The authors have answered all my inquiries and included their comments to the original manuscript. The supplemental material adds valuable information for the reader.